# A Genome-Wide Association Study of Oxypurinol Concentrations in Patients Treated with Allopurinol

**DOI:** 10.3390/jpm14060649

**Published:** 2024-06-18

**Authors:** Maxime Meloche, Marc-Olivier Pilon, Sylvie Provost, Grégoire Leclair, Essaïd Oussaïd, Isabelle St-Jean, Martin Jutras, Marie-Josée Gaulin, Louis-Philippe Lemieux Perreault, Diane Valois, Ian Mongrain, David Busseuil, Jean-Lucien Rouleau, Jean-Claude Tardif, Marie-Pierre Dubé, Simon de Denus

**Affiliations:** 1Faculty of Pharmacy, Université de Montréal, Montreal, QC H3T 1J4, Canada; 2Montreal Heart Institute, Montreal, QC H1T 1C8, Canadadavid.busseuil@icm-biobanque.org (D.B.);; 3Université de Montréal Beaulieu-Saucier Pharmacogenomics Centre, Montreal, QC H1T 1C8, Canada; 4Faculty of Medicine, Université de Montréal, Montreal, QC H3T 1J4, Canada

**Keywords:** GWAS, pharmacogenomics, allopurinol, drug metabolism, biobank

## Abstract

Cohort studies have identified several genetic determinants that could predict the clinical response to allopurinol. However, they have not been commonly used for genome-wide investigations to identify genetic determinants on allopurinol metabolism and concentrations. We conducted a genome-wide association study of a prior cross-sectional investigation of patients from the Montreal Heart Institute Biobank undergoing allopurinol therapy. Four endpoints were investigated, namely plasma concentrations of oxypurinol, the active metabolite of allopurinol, allopurinol, and allopurinol-riboside, as well as allopurinol daily dosing. A total of 439 participants (mean age 69.4 years; 86.4% male) taking allopurinol (mean daily dose 194.5 mg) and who had quantifiable oxypurinol concentrations were included in the genome-wide analyses. Participants presented with multiple comorbidities and received concomitant cardiovascular medications. No association achieved the predefined genome-wide threshold values for any of the endpoints (all *p* > 5 × 10^−8^). Our results are consistent with prior findings regarding the difficulty in identifying genetic determinants of drug concentrations or pharmacokinetics of allopurinol and its metabolites, as well as allopurinol daily dosing. Given the size of this genome-wide study, collaborative investigations involving larger and diverse cohorts may be required to further identify pharmacogenomic determinants of allopurinol and measure their clinical relevance to personalize allopurinol therapy.

## 1. Introduction

Gout is characterized by inflammatory arthritis that arises from deposition of urate crystals in joints following persistent hyperuricemia [1]. This causes recurrent painful flares while increasing the risk of permanent joint damage and disability. Over a prolonged timeframe, elevated serum uric acid (SUA) concentrations have been shown to be a significant risk factor correlating with the development of numerous cardiovascular, renal, and metabolic disorders [2,3,4]. To normalize SUA levels, guidelines from rheumatology associations recommend allopurinol, a purine analog and xanthine oxidase (XO) inhibitor, as first-line urate-lowering therapy for the management of chronic gout [5,6,7]. In the liver, allopurinol is rapidly and extensively metabolized into the active metabolite oxypurinol, while a minor fraction undergoes conjugation into allopurinol-riboside [8,9]. The greater pharmacological potency and half-life of oxypurinol relative to allopurinol (~24 h vs. ~1.5 h) are responsible for most of the urate-lowering effect by limiting uric acid synthesis through the XO pathway [9,10]. Oxypurinol is almost exclusively excreted through the kidney in its unchanged form [11,12], thus making renal function an important consideration prior to administration [5].

Despite the frequent use of allopurinol in the clinic, previous literature has acknowledged the substantial variability in clinical response. General factors have been highlighted, such as the poor treatment adherence that often prevents patients from reaching therapeutic target SUA thresholds [13,14,15]. Recent efforts have also suggested the inclusion of clinical parameters alongside renal function for optimizing current dosing algorithms and accounting for interindividual differences [16,17,18,19]. Genome-wide association studies (GWASs) specific to allopurinol therapy have historically allowed the discovery of single-nucleotide polymorphisms (SNPs) as risk factors for life-threatening cutaneous reactions [20,21], resulting in routine clinical testing being implemented [22,23]. They have also been successfully applied to detect genetic risk factors associated with baseline SUA concentration levels and assist in predicting clinical outcomes in gout [24,25,26,27,28]. More recently, novel pharmacogenomic (PGx) determinants have been associated to allopurinol clinical response across different populations. Both candidate association studies and GWASs have repeatedly implicated variants located within genes coding for common gut and renal transporters. One example includes *ABCG2* coding for the breast cancer resistance protein (BCRP) for which urate, allopurinol, and oxypurinol are substrates [28,29,30,31,32]. Interestingly, GWA analyses further revealed a novel SNP in *GREM2* associated with responder rates but not baseline SUA levels, implying that unsuspected genomic regions may harbor PGx elements strictly affecting allopurinol pharmacodynamics instead of physiologic urate transport [33].

Past PGx investigations on drug concentrations have usually centered around known biological targets. However, the capabilities of GWA methodologies have been demonstrated in cohort studies to detect new PGx elements of drug concentrations with smaller effect sizes beyond well-established transporter and enzymatic pathways [34,35]. To our knowledge, this approach has not been performed to investigate variations in allopurinol metabolism. With the resources that institutional biorepositories provide, we sought to utilize similar methods and assess whether randomly collected biobank samples could be leveraged to detect PGx determinants of plasma allopurinol and metabolite concentrations, together with daily dosing.

## 2. Materials and Methods

### 2.1. Study Design and Participant Selection

Participants from the previous cross-sectional study were selected from the MHI Biobank’s records. The methods of the MHI Biobank, including patient enrollment and sample storage protocols, have been described in prior works [36,37]. Briefly, the biobank consists of biological material and encoded data from patients of the MHI Hospital Cohort who have used the hospital center’s services and provided informed consent for their participation. For this study, selected records and biological samples were retrieved from patients who had enrolled in the MHI Biobank with baseline values taken between 22 May 2007 and 12 September 2018. Samples were acquired from patients randomly with regards to allopurinol intake, concomitant medications, time of day, or food intake. Along with biological sampling, questionnaires were administered to participants concerning personal and family medical history, as well as individual pharmacological, dietary, and psychosocial data, among others. Except for plasma concentrations of allopurinol and its metabolites, all information was collected directly from participants, their medical records, and MHI electronic databases. No prospective recruitment was performed as part of this investigation.

As in our initial cross-sectional study, only self-reported “White” males and females aged ≥ 18 years were included to minimize confounding risks and population stratification [38]. Selected participants were treated with allopurinol and needed to have had plasma samples available at the time of their enrolment. In the current study, for 24 patients, plasma was collected during follow-up, at which time a full pharmacological and medical history was again completed and used as part of these analyses. Considering the extended half-life for oxypurinol, patients with non-quantifiable concentrations of the metabolite were deemed non-adherent to treatment and excluded from all analyses. We also excluded participants with a history of heart, kidney, or liver transplant since the genotype from the donor and the recipient could differ, thus removing recipient–donor interactions that could mislead inferences on a patient’s metabolic capacity [39,40].

### 2.2. Study Endpoints 

The endpoints investigated in the initial study were plasma oxypurinol, allopurinol, and allopurinol-riboside concentrations, as well as patients’ daily allopurinol dosing [38]. One patient had missing allopurinol dose values and was removed from the final analyses. All concentration levels were measured from blood samples obtained during each participant’s baseline visit. 

### 2.3. Quantification of Oxypurinol, Allopurinol, and Allopurinol-Riboside Plasma Concentrations

The quantification of oxypurinol, allopurinol, and allopurinol-riboside was performed from plasma samples collected upon enrolment in the MHI Biobank. All analyses were conducted at the bioanalytical laboratory of the Platform of Biopharmacy at the Université de Montréal. Details of the full bioanalytical method have been described previously [38]. Briefly, oxypurinol, allopurinol, and allopurinol-riboside analyses were carried in a blinded fashion using a high-pressure liquid chromatography system coupled with electrospray ionization tandem mass spectrometry and based on selective multiple reaction monitoring. Validated quantification ranges were 10–50,000 ng/mL for all analytes. Concentrations below the lower limit of quantification (LLOQ < 10 ng/mL) were given a zero value as part of the analyses. 

### 2.4. Genotyping Quality Control and Imputation

The genotyping quality control and imputation methods, as detailed in previous works [41], were employed in this study. Genome-wide genotyping was performed using 200 ng of genomic DNA at the Beaulieu-Saucier Pharmacogenomics Centre (Montreal, QC, Canada). The Illumina Infinium Global Screening Array v3-MD (Illumina, CA, USA) was utilized following the manufacturer’s instructions. BeadChips were subsequently scanned using the Illumina iScan, with data analysis carried out using the data manifest MHI_GSAMD-24v3-0-EA_20034606_C1.bpm. Plink files were generated with the iaap-cli tool (version 1.1.0–80d7e5b). Intensities, B allele frequency, and log R ratio were extracted using the gtc_convert tool (version 0.1.2). Quality control and genetic data cleanup procedures were performed using PyGenClean (version 1.8.3) [42] and PLINK (versions 1.07 and 1.9; the latter for the data manipulation steps of the relatedness and ethnicity modules) [43].

The genotyping experiment involved 184 plates of DNA samples. One control was added per hybridization run, which corresponded to two plates, and was chosen from NA06994, NA12717, NA12878, NA18861, and NA19147 that were obtained from the National Institute of General Medical Sciences Human Genetic Cell Repository at the Coriell Institute for Medical Research. The completion rate threshold for genotypes and samples was set to 99%. Cryptic relatedness among samples and sample outliers were identified using the pairwise identity by state matrix and multidimensional scaling, respectively. The first two multidimensional scaling components of each subject were plotted, including the genotypes of HapMap CEU, JPT-CHB, and YRI reference samples. Outliers from the CEU cluster were flagged and removed by k-nearest neighbor with a threshold of 1.9σ in PyGenClean (version 1.8.3). Principal components were computed for the selected study samples to account for population structure [44]. We excluded nine participants who were not of European ancestry. Genome-wide imputation was carried out with the TOPMed Imputation Server (version 1.5.7) [45] using Eagle (version 2.4) [46] for phasing and Minimac4 (version 1.0.2) [45] for imputation. Gene information was retrieved with Annovar (version 2020-06-07). The rsID numbers were retrieved with Ensembl REST API. Genetic variants from pseudo-autosomal regions were analyzed as autosomal variants, with all positions obtained from build 38. A total of 61,640,018 genetic variants with a quality value (r2) ≥ 0.6 remained, of which 6,394,414 had a minor allele frequency of ≥5%.

### 2.5. Statistical Analyses 

Descriptive statistics were obtained on the cohort’s demographics, clinical characteristics, and genotype information. For all parameters, means and standard deviations were used for reporting continuous variables, whereas counts with percentages were used for categorical variables. To satisfy normality assumptions, outcomes were log-transformed, and the distributions of the residuals were used. Models with transformed outcomes all resulted in more normally distributed measures relative to untransformed datasets.

The GWAS analyses were conducted using the Scalable and Accurate Implementation of GEneralized mixed model (SAIGE) version 0.44.6.5 on R software (The R Foundation for Statistical Computing) [47]. This package uses a linear mixed regression framework with individual-level data to analyze large-scale datasets while simultaneously controlling for sample relatedness and case–control imbalances. Sample relatedness was accounted for with the use of a genetic relationship matrix for random effects. Phenotype and genotype data were merged for all participants, and only those with European ancestry, related or not, were included. Multivariable linear regression modeling was performed for every outcome assessed, with covariable selection made to maximize convergence: plasma oxypurinol concentrations were adjusted using age, sex, weight, daily allopurinol dosage, and ten principal components (PC1-10); plasma allopurinol and allopurinol-ribose concentrations were both adjusted using age, sex, daily allopurinol dosage, and PC1-5; allopurinol dosage was adjusted using age, sex, and PC1-5. Then, as part of exploratory analyses, we used the same regression models to investigate variants from multiple GWASs that were previously associated with SUA concentrations and allopurinol response in large cohorts [25,26,33]. All genetic variants with significance threshold *p* < 10^−4^ were compiled into the final GWA results. For every statistical test, the significance threshold was set at 5 × 10^−8^.

## 3. Results

### 3.1. Study Cohort

Overall, the final GWA analysis consisted of 439 participants with quantifiable plasma oxypurinol concentrations, with baseline patient characteristics consistent relative to those from the initial cohort treated with allopurinol (Table 1) [38]. Patients were prescribed allopurinol daily doses averaging 194.5 mg ± 77.1 (range 42.9 (100 mg 3 times per week) to 600 mg). As expected, patients presented with comorbidities and received various concomitant cardiovascular drugs.

### 3.2. Genome-Wide Association Analyses: Allopurinol Metabolism and Dosing

There were no variants that reached significance thresholds for either oxypurinol, allopurinol, or allopurinol-ribose plasma concentrations (Figure 1a–c). After multivariable regression modeling, variants that showed the strongest associations were located predominantly in intergenic regions, followed by intronic portions of non-coding RNA (ncRNA) (Appendix A). More precisely, plasma oxypurinol concentrations had the highest association with the intergenic variant chr5:98238797:G:A closest to *LINC01846* (*p* = 2.9 × 10^−5^). Meanwhile, plasma allopurinol and allopurinol-riboside concentrations showed the greatest significance with intergenic variants chr6:49561859:T:C (*p* = 1.3 × 10^−6^) and chr7:8405423:G:A with nearest proximity to *C6orf141* and *NXPH1* (*p* = 6.5 × 10^−7^), respectively.

Regarding daily allopurinol dosing, the strongest associations were detected within intronic ncRNA regions of the *LINC02588* gene, although none reached the predefined significance thresholds (all *p* > 2.9 × 10^−7^) (Figure 1d). Additional signals were found in intronic regions of two more loci, namely *HLA-DQB1* (chr6:32663671:A:G; *p* = 9.4 × 10^−7^) and *DNAJC25-GNG10* (rs1570303; *p* = 9.2 × 10^−7^).

Results for the lookups of previously identified genetic variants of allopurinol response showed no variant reaching statistical significance thresholds from the previously reported genome-wide analyses in our cohort (Appendix A). This could further underscore the likely role of these SNPs in regulating either physiologic SUA concentrations, gout risk, and allopurinol/oxypurinol pharmacological effect rather than their pharmacokinetic profiles or drug concentrations.

## 4. Discussion

Allopurinol is a mainstay of gout therapy due to its demonstrated urate-lowering benefits, well-tolerated safety profile, and high affordability [2,3,4,48]. Studies using candidate gene and genome-wide approaches have revealed numerous SNPs of membrane transporters that may predict clinical response to allopurinol. In this study, we conducted a GWAS to identify PGx determinants on allopurinol concentrations and metabolism. We found no association that reached statistical significance regarding variants that would affect the concentration levels of allopurinol and its metabolites. The top SNP of our GWA analyses of plasma oxypurinol concentrations was a variant located in an intergenic region near the *LINC01846* gene. A long intergenic ncRNA, there is little information available regarding its biological function. Whether nearby genomic alterations or interactions can impact its role in pharmacokinetics or drug concentrations more broadly, if any, has yet to be elucidated. Allopurinol concentrations had a top SNP closest to the *C6orf141* gene. An open reading frame with higher relative expression in the duodenum and gallbladder, the function of *C6orf141* has not been extensively studied, although cohort data indicate a tumor-suppressor effect in squamous cell oral cancer [49]. As for plasma allopurinol-riboside concentrations, we also observed intergenic variants that showed near significance close to *NXPH1*. Altogether, even if associations of varying significance have been made in prior GWASs with a wide range of phenotypes for those genes [50], we failed to demonstrate statistical significance. Their involvement in treatment response variability and pharmacokinetics therefore requires more extensive evidence.

We found suggestive association signals in regions of the long ncRNA transcript *LINC01588* with daily allopurinol dosing. Similar to genetic elements associated to drug concentrations, functional data on this gene are scarce, thus making any plausible biological or pharmacological implication uncertain. However, it is worth noting that additional significant genomic associations were also seen with the well-characterized *HLA-DQB1* gene. Different from the variant we detected, GWAS results from cross-ancestral meta-analyses identified SNPs nearby *HLA-DQB1* associated to SUA concentrations [51,52]. Though unclear, those impacts from genetic variations could signify that the HLA region may regulate the renal transport of uric acid or other urate-dependent inflammatory mechanisms, therefore impacting gout susceptibility and staying in line with its roles in immunity and inflammation [51,53,54]. Again, further investigations are required to support these findings. 

Studies evaluating allopurinol and oxypurinol pharmacokinetics have been unable to consistently identify PGx determinants [31,55]. Even recently, a cohort of 300 gout patients was used to predict the impacts of genetic variability from multiple gut and renal urate transporters on oxypurinol metabolism through population pharmacokinetic modeling [56]. After assessing several known variants, no signal was detected for any of the membrane transporter genotypes when accounting for common clinical variables. As previously emphasized, increasing evidence suggests that allopurinol metabolism is influenced by a combination of factors including oligogenic predictors, clinical and anthropometric variables, SUA regulation, and endogenous markers [28,29,57,58]. Thus, varied interactions between these parameters could define allopurinol and metabolite pharmacokinetics, as well as its response, with the involvement of pathways that extend beyond those impacting drug concentration levels [26,33]. This is illustrated by the recent discovery of unknown mechanisms of action of oxypurinol in gout. GWA analyses followed by joint functional assays were able to validate the uricosuric properties of oxypurinol through direct inhibition of GLUT9-mediated urate reabsorption [26,33]. Still, one could assert that the pleiotropic effects of BCRP in gout, and potentially other membrane transporters, would rather arise from a modulation of allopurinol pharmacodynamics and urate reuptake than changes in the drug’s pharmacokinetics, thus justifying the lack of significant metabolic variability across investigations.

### Study Limitations

Our study does contain some limitations, the size of our study population being an important one. Previous observational cohorts with subsequent meta-analyses have been performed for replication of previous observations with *ABCG2* and *GREM2* [29,33]. In the future, this could imply that multiple observational cohorts are needed through collaborative efforts in assessing genetic predictors of allopurinol metabolism and concentrations on a genome-wide scale. To this end, our GWAS results have been made publicly available via the PheWeb portal (https://pheweb.statgen.org/allo-mhi/), which could enable meta-analyses to be carried out, therefore overcoming the limitations of small-scale studies. The composition of our cohort may also limit the interpretation of this study. For example, >26% of participants presented with chronic heart failure, a factor known for influencing the clinical pharmacokinetics and metabolism of drugs [59]. Although this does not invalidate our results, it is possible that allopurinol and metabolite concentrations could have been affected by unaccounted-for disease-induced physiological changes. Furthermore, we limited our investigation to participants of European ancestry. As was recently demonstrated, multi-ethnic cohorts have allowed the identification and validation of PGx signals impacting both plasma drug concentrations and metabolic ratios while displaying consistent effect sizes across ancestries [60]. Therefore, limiting the study to a single ancestry may have hindered the detection of PGx variants impacting allopurinol and metabolite concentrations. Finally, since participants from the MHI Biobank had blood sampling performed without consideration to timing post-dose, adequate quantification of drugs with significantly shorter half-lives may prove to be challenging using a random sampling approach. In our case, in which oxypurinol had a longer half-life and was the primary analyte of interest, such limitation did not affect our main objective but could prevent allopurinol from being reliably quantified across cohorts taking the drug.

## 5. Conclusions

In conclusion, the current GWAS did not identify PGx determinants associated to plasma concentration levels of oxypurinol, allopurinol, and allopurinol-riboside, as well as daily allopurinol dosing. These results support the notion that complex gene interactions, non-genomic markers, or multiple PGx elements with modest effect sizes may be identified as collaborative investigations expand to include larger patient cohorts. Ultimately, more definitive PGx signals regarding allopurinol pharmacokinetics and clinical response might be ascertained in future works to predict clinical outcomes in gout patients.

## Figures and Tables

**Figure 1 jpm-14-00649-f001:**
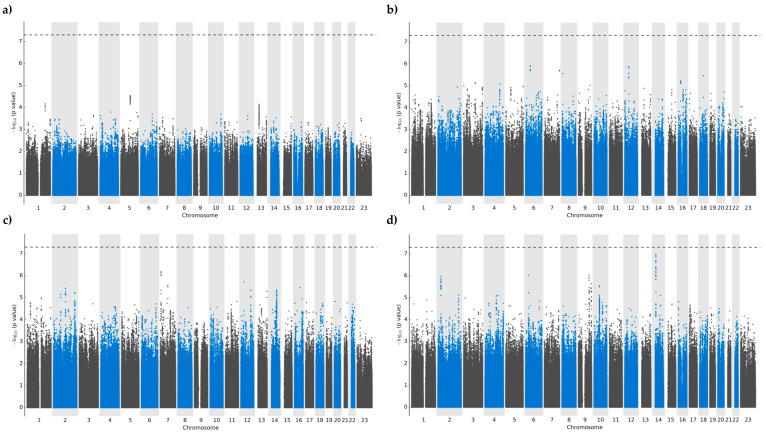
Manhattan plot of genome-wide markers assessed for log-transformed plasma concentrations for (**a**) oxypurinol, (**b**) allopurinol, (**c**) allopurinol-riboside, and (**d**) daily allopurinol dose in 439 European participants. Dotted lines represent a significance threshold of 5 × 10^−8^.

**Table 1 jpm-14-00649-t001:** Baseline cohort characteristics.

Characteristics	*n* = 439 (100%) ^1^
Age (years)	69.4 (8.0)
Females, *n* (%)	64 (14.6)
Smoking status, *n* (%)	
Never-smoker	117 (26.7)
Past-smoker	301 (68.6)
Current-smoker	21 (4.8)
Weight (kg)	90.1 (18.3)
BMI	31.4 (5.5)
Hypertension, *n* (%)	379 (86.3)
Diabetes mellitus, *n* (%)	
Type 1	1 (0.2)
Type 2	182 (41.5)
Dyslipidemia, *n* (%)	383 (87.6)
Myocardial infarction, *n* (%)	173 (39.7)
Chronic heart failure, *n* (%)	113 (25.9)
Chronic renal failure, *n* (%)	115 (26.2)
Analyte concentrations	
Mean daily allopurinol dose (mg)	194.5 (77.1)
Mean quantifiable oxypurinol plasma concentrations (ng/mL)	13,374.4 (8,656.6)
Mean allopurinol plasma concentrations (ng/mL)	277.6 (358.1)
Mean allopurinol-riboside plasma concentrations (ng/mL)	228.3 (206.3)
Concomitant medications, *n* (%)	
Aspirin	307 (70.1)
Other antiplatelet agents	62 (14.2)
ACE inhibitors	159 (36.2)
Angiotensin II receptor blockers	172 (39.2)
Beta-blockers	315 (71.8)
Calcium channel blockers	149 (33.9)
Amiodarone	20 (4.6)
Warfarin	116 (26.4)
Novel oral anticoagulants	19 (4.3)
Digoxin	54 (12.3)
Diuretics	263 (59.9)
Statins	356 (81.1)
Fibrates	16 (3.6)
Other hypolipidemic agents	53 (12.1)
Oral hypoglycemic agents	159 (36.3)
Insulin	34 (7.7)
Serum creatinine (*n* = 391, 89.1%)	
Concentrations (µmol/L)	118.7 (54.5)

Abbreviations: ACE, angiotensin-converting enzyme; BMI: body mass index. Values are presented as means (standard deviation) unless otherwise stated. ^1^ Numbers are rounded to the first decimal.

## Data Availability

The original data presented in the study are openly available in the PheWeb online repository at https://pheweb.statgen.org/allo-mhi/.

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
