# Peer review of "A Genome-Wide Association Study of Oxypurinol Concentrations in Patients Treated with Allopurinol"

_jpm, 2024, doi:10.3390/jpm14060649_

Round 1

Reviewer 1 Report

Comments and Suggestions for Authors

Dear Authors,

According to the guidelines for reviewers, I would like to report on my reviewing progress.

The main question addressed by this study is to test whether the genome-wide association study (GWAS) of a prior cross-sectional investigation of patients from the Montreal Heart Institute Biobank undergoing allopurinol therapy may enable to detect PGx determinants of plasma allopurinol and metabolite concentrations, together with daily dosing

This study aimed to apply and test the capabilities of GWA methodologies in cohort study to detect new PGx elements of drug concentrations with smaller effect sizes beyond well-established transporter and enzymatic pathways, since this approach has not been performed to investigate variations in allopurinol metabolism.

This study adds to the subject area compared with other published material the results of the examination of the four endpoints that were investigated (plasma concentrations of oxypurinol, the active metabolite of allopurinol, allopurinol, and allopurinol-riboside, as well as allopurinol daily dosing), although no association achieved the predefined genome-wide threshold values for any of the endpoints (all p>5 × 10-8).

The topic is original and relevant in the field, since despite the frequent use of allopurinol in the clinic, there is a substantial variability in clinical response. In addition, cohort GWA studies are useful in attempt to detect new PGx elements of drug concentrations with smaller effect sizes beyond well-established transporter and enzymatic pathways.

The type of article is well selected, since the Articles are original research manuscripts that include the most recent and relevant references in the field.

The title is informative and it relates to the content of the article. However, the end of title requires the addition “an observational cohort study”.

Keywords are appropriate and reflect the content of the article.

The Abstract is structured, the background, methods, results and conclusion are provided in Abstract.

In Introduction section, the authors explained the background of the problem, relevant literature is listed and hypothesis is clearly elaborated. The study has a novelty. In the Introduction section, the authors described in a clear and unambiguous manner their aim at the end of the Introduction.

The Material and methods selected are adequate and precisely described. Study design and participant selection is adequately described. The study endpoints are precisely defined. The methods for drugs plasma concentrations are adequate, as well as genotyping. However, in Statistical analysis, the conventional genome-wide significance p value threshold of 5 × 10-8 is neither mentioned nor its selection described (this threshold is mentioned only twice within the lines 22 and 215 of manuscript, but it is also required in Materials and Methods section of manuscript).

Results of the study are presented in a clear manner in 1 Figure and 8 supplemental tables that accurately represent the results. 

Scientific content of this article is relevant and well presented. In Results section, authors fulfilled in adequate manner the goal of their GWAS to identify PGx determinants on allopurinol concentrations and metabolism and showed that no association was found that reached statistical significance regarding variants that would affect the concentrations levels of allopurinol and its metabolite

In interpreting the results in Discussion section, the previous research by the authors and others has been discussed and those results are compared to the current results. Numerous potential limitations of the present study are described in details. 

The Conclusions are consistent with the evidence and arguments presented and they address the main question posed.

The references are appropriate, they are up-to-date and numbered consecutively in the manuscript. However, one of the key articles in this domain has not been cited (Lancet. 2008; 372(9654):1953-61.), and its importance is reflected by the fact that it is the first publication in which ABCG2 and SLC17A3 are described as new loci associated with gout in Caucasian and African-American cohort.

Reviewer 2 Report

Comments and Suggestions for Authors

Fascinating and extensive study. I have only a few questions about why only males are highlighted in the abstract. When I look at suppl. data and found lots of SNP are lying in a gene. If we analyzed Haplotypes and Gene-Gene interaction, the author will get a good risk score. 
